# Towards Finding Longer Proofs

## Abstract

We present a reinforcement learning (RL) based guidance system for automated theorem proving geared towards Finding Longer Proofs (FLoP). FLoP focuses on generalizing from short proofs to longer ones of similar structure. To achieve that, FLoP uses state-of-the-art RL approaches that were previously not applied in theorem proving. In particular, we show that curriculum learning significantly outperforms previous learning-based proof guidance on a synthetic dataset of increasingly difficult arithmetic problems.

## 1 Introduction

In 1958 B. F. Skinner, a pioneer of modern behaviorism, in the article "Teaching Machines" (Skinner, 1958) noticed that "in acquiring complex behavior the student must pass through a carefully designed sequence of steps, often of considerable length. Each step must be so small that it can always be taken, yet in taking it the student moves somewhat closer to fully competent behavior". His study extended also to the teaching of arithmetic: "The student is expected to arrive at $9 \cdot 7 = 63$, not by memorizing it as he would memorize a line of poetry, but by putting into practice such principles as that nine times a number is the same as ten times the number minus the number ...". The idea of learning using a curriculum of problems is also widely used in machine learning (Bengio et al., 2009; Elman, 1993; Resnick et al., 2018; Salimans & Chen, 2018) and in this work we apply curriculum learning to automatic theorem proving focusing on arithmetic.

Our work has the following contributions. (1) We introduce a new theorem proving algorithm FLoP (Section 4) based on reinforcement learning and the connection tableau calculus. FLoP uses a meta-learning variation of the curriculum learning algorithms presented by Resnick et al. (2018) and Salimans & Chen (2018). (2) We introduce a synthetic dataset of increasingly difficult arithmetic problems organized as RL environments (Section 5). (3) We use this benchmark to compare (Section 6) the performance of our system with state-of-the-art saturation provers Vampire (Kovács & Voronkov, 2013) and E (Schulz, 2013) guided by human-designed strategies, and with rlCoP (Kaliszyk et al., 2018) – a recently developed RL-based connection tableau prover. FLoP significantly outperforms the other provers on harder problems, demonstrating its ability to find longer proofs.

Our datasets presented in Section 5 seem to be particularly suited for machine learning methods: problems are simple, solutions are long, repetitive and rather predictable for humans. Still, state-of-the-art systems struggle with solving some of the problems – see Section 6 for details.

Other works using machine learning to guide a prover (Chvalovský et al., 2019; Jakubuv & Urban, 2017; Kaliszyk & Urban, 2015b; Kaliszyk et al., 2018; Loos et al., 2017; Urban et al., 2011) usually deal with large mathematical corpora, while we focus on a fragment of Robinson Arithmetic, which is a limited and simple theory. Our reasons behind this narrower focus: (a) We wanted to create a scalable RL benchmark with emphasis on the length of proofs. (b) A symbolic method based on human-designed sets of hints (Veroff, 1996) was previously successfully applied in abstract algebra by Kinyon et al. (2013) to discover long proofs and we wanted to check whether learning of long proofs is feasible using the state-of-the-art ML toolset. (c) We wanted interpretable failure modes. In the case of large mathematical corpora, the interpretation of failures may be a hard task because of multiple failures and the complicated structure of the corpora, requiring specialized domain knowledge both in mathematics and with regard to the inner workings of the proof system.

Our code, datasets and all experiment configuration files are available at `http://bit.ly/code_atpcurr`[1]. Supplementary materials including screencasts with gameplays performed in our environments are available at the project webpage `http://bit.ly/site_atpcurr`.

## 2 RELATED WORK

**Machine learning datasets and RL environments involving mathematics and logic.** The arithmetic dataset which we introduce in Section 5 is geared towards longer proofs and is structurally much simpler than other theorem proving datasets which we list below. One can think about this suite of RL problems as gridworlds of theorem proving (see (Sutton & Barto, 2018, Example 3.5) for a broader explanation of importance of gridworlds in RL). Our dataset is intended to become a general purpose testing ground for theorem proving and reinforcement learning methods, in particular for meta-learning and hierarchical learning algorithms.

TPTP (Sutcliffe, 2017) consists of 22507 problems in 53 domains collected over several decades. A large dataset for developing machine learning for theorem proving based on the Mizar Mathematical Library (MML) (Grabowski et al., 2010) was introduced by Urban (2006a) in the MPTP project. The dataset was used e.g. by Alemi et al. (2016); Kaliszyk & Urban (2015a); Urban (2007); Urban et al. (2008). Similar datasets based on the Isabelle/HOL, HOL Light/Flyspeck and HOL4/CakeML systems and projects (Blanchette et al., 2016; Gauthier & Kaliszyk, 2015; Kaliszyk & Urban, 2014) were introduced in the last decade and used for the CASC LTB (large theory) ATP competition (Sutcliffe & Urban, 2016) and other system evaluations. Such datasets cover large areas of mathematics and computer science and contain diverse axioms, lemmas, theorems, definitions, and symbols. Smaller subsets of lemmas leading to the Bolzano-Weierstrass theorem were selected from the MPTP dataset to form the MPTP Challenge (Urban, 2006b) and the MPTP2078 benchmark (Alama et al., 2014). HOLStep (Kaliszyk et al., 2017) introduced a dataset based on 11400 proofs, including a proof of the Kepler Conjecture (Hales et al., 2015), formalized using HOL Light (Harrison, 1996). In HOLStep and in FormulaNet (Wang et al., 2017) the dataset was used as a benchmark for various neural architectures. The recent HOList project (Bansal et al., 2019a;b; Paliwal et al., 2019) uses 29462 theorems formalized in HOL Light and instruments them for experiments oriented towards tactic selection, where a tactic is a human-designed program which aggregates multiple proof steps. GamePad (Huang et al., 2019) introduced a dataset based on a formalization of the Feit-Thompson Theorem (Gonthier et al., 2013) along with a generated algebra problems. It is intended for learning tactic selection together with an auxiliary task of predicting the number of proof steps left. A dataset based on theorems proved in HOL4 (Slind & Norrish, 2008) was used for developing the TacticToe (Gauthier et al., 2018) learning-guided tactical prover. Saxton et al. (2019) proposed a dataset of simple algebraic problems expressed in English. Arithmetic problems without a natural language context were tackled by Neural GPUs (Kaiser & Sutskever, 2016) and its improved successors (Freivalds & Liepins, 2017). Supervised learning was also applied to various instances of propositional satisfiability in NeuroSAT (Selsam et al., 2018) and NeuroCore (Selsam & Bjørner, 2019) as well as in (Chvalovsky, 2019; Evans et al., 2018). Datasets introduced in Section 5 are OpenAI-gym (Brockman et al., 2016) compliant and can be tested with modern RL algorithms. Previous work on theorem proving and RL includes TacticToe, HOList and rlCoP (Kaliszyk et al., 2018). TacticToe and rlCoP use guided Monte Carlo Tree Search (MCTS) and HOList proposes a custom RL algorithm.

**Machine learning systems for guidance of theorem provers.** Our current work focuses on providing guidance for the fCoP (Kaliszyk et al., 2015b) theorem prover. fCoP is an OCaml implementation of the very compact lean connection tableau prover (Otten & Bibel, 2003). fCoP was used as the proof engine in the guided provers FEMaLeCoP (Kaliszyk & Urban, 2015b) and rlCoP (Kaliszyk et al., 2018). FEMaLeCoP learns only from positive data using two simple, but fast machine learning models (custom nearest neighbour and naive Bayes). In rlCoP, the value and policy functions of the guided MCTS algorithm are learned similar to (Anthony et al., 2017; Silver et al., 2017), using gradient boosted trees as implemented in the XGBoost (Chen & Guestrin, 2016) library. In contrast, we use neural network models instead of trees and the Proximal Policy Optimization (PPO) (Schulman et al., 2017) algorithm instead of MCTS. In a longer run we believe that these methods should be combined, see (François-Lavet et al., 2018, Section 6.2), but in this work we

---

[1]This distribution does not include the fCoP theorem prover, which cannot yet be publicly released, however a binary can be obtained upon request.

propose to investigate how much can be achieved directly via rollouts and without a search algorithm like MCTS. A distinctive feature of our approach is the ability to perform very long rollouts both in training and evaluation. We demonstrate this in Section 6, see Figure 4. Chvalovský et al. (2019); Jakubuv & Urban (2017; 2019); Loos et al. (2017) added learning-based guidance to E prover (Schulz, 2013). These are supervised experiments which learn from saturation-style proof traces.

**Meta-learning suites of RL environments.** Meta-learning algorithms in the context of RL can be tested using a suite of simulated robotic tasks (Finn et al., 2017; Rakelly et al., 2019), one of discrete environments proposed in (Cobbe et al., 2018; Juliani et al., 2019; Menashe & Stone, 2018; Nichol et al., 2018; Roderick et al., 2017) or a new MineRL (Guss et al., 2019) suite of problems with mixed continuous and discrete actions. Our suite of tasks involves discrete actions.

**Curriculum learning and reinforcement learning.** Our algorithm FLoP is an adaptation of the curriculum learning algorithms presented in (Resnick et al., 2018; Salimans & Chen, 2018) in a context of a suite of reinforcement learning environments presented in Section 5.

## 3 FCoP AND THE CONNECTION TABLEAU CALCULUS

In this section, we give a brief overview of the connection tableau method, as implemented by the fCoP system. We assume basic first-order logic and theorem proving terminology (Robinson & Voronkov, 2001). The input is a (mathematical) problem consisting of *axioms* and *conjectures* formally stated in first-order logic (FOL). The calculus searches for *refutational proofs*, i.e. proofs showing that the axioms together with the negated conjectures are *unsatisfiable*. The FOL formulas are first translated to *clause normal form* (CNF), producing a set of first-order *clauses* consisting of *literals* (atoms or their negations). Figure 1 shows a *closed connection tableau*, i.e., a finished proof tree where every branch contains *complementary literals* (literals with opposite polarity). Since all branches contain a pair of contradictory literals, this shows that the set of clauses is unsatisfiable. Proof search starts with a *start clause* as a *goal* and proceeds by building a connection tableau by repeatedly applying *extension steps* and *reduction steps*.

The extension step connects (*unifies*) the *current goal* (a selected tip of a tableau branch) with a complementary literal of a new clause. This extends the *current branch*, possibly splitting it into several branches if there are more literals in the new clause, and possibly *instantiating* some variables in the tableau. The reduction step connects the current goal to a complementary literal of the *active path*, thus *closing* the current branch. The proof is finished when all branches are closed. The extension and reduction steps are nondeterministic, requiring backtracking in the standard connection calculus. Brute force search such as *iterative deepening* can be used to ensure completeness, i.e., making sure that the proof search finds a proof if there is any.

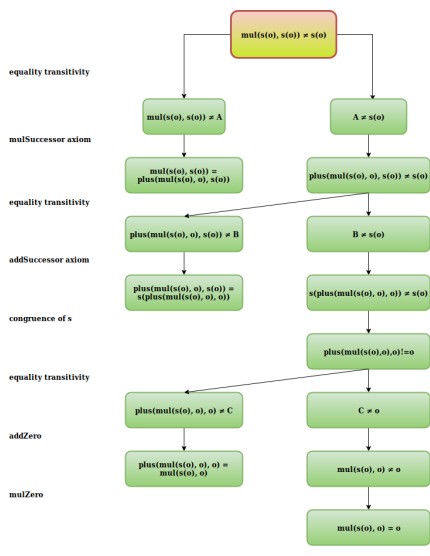

Figure 1: *A closed tableau tree of the proof of* $1 \cdot 1 = 1$. *On the left we list the actions taken in the proof. See* http://bit.ly/site_atpcurr *for details.*

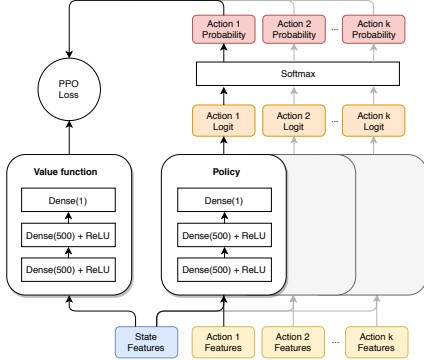

Figure 2: *Value and Policy network architectures in PPO. Their inputs are state and state-action pair features, respectively. The policy returns a score for each action, which are then normalized to a probability.*

fCoP represents theorem proving as a one-person game. The game ends with a success if a proof is found. The prover has many choices to make along the way, hence it typically has to explore a search space that is exponentially large in the length of the proof. In fCoP, the action space is roughly correlated with the size of the axiom set. While this can be large for large problems, typically only a few actions are available in any particular state.

## 4  FLoP – the Main Algorithm

The FLoP algorithm combines the connection tableau calculus with guidance based on PPO and curriculum learning Resnick et al. (2018); Salimans & Chen (2018). Actions in our theorem proving game consist of selecting an extension step as defined in Section 3 (reduction steps are performed automatically by the game engine). Figures 2 and 3 show how actions interact with other components of FLoP. Each extension step involves selecting one of the clauses, however, not all clauses are applicable as actions at a given proof step, due to the unification condition. The full information about the game state consists of all previous proof steps, the partial proof tree (proof state) and the current goal. The state and actions (formulas) are represented using previously developed features Kaliszyk & Urban (2015b); Kaliszyk et al. (2015a; 2018). The features mainly include (suitably hashed) triples of adjacent nodes in the formula trees and in the partial proof trees. This means that the proof states and the actions are presented as (sparse) fixed length vectors, see the inputs to the policy and value networks in Figure 2. These features have proved useful but are not free from problems. See the discussion in Appendix A.

**Curriculum Learning on Proofs.** In theorem proving we are dealing with sparse rewards and we tackle this with the help of curriculum learning as implemented in Algorithm 1.

First, in line 6 of Algorithm 1 we sample a problem. In lines 7-20 we play an episode: if we have a proof then we start from the state dictated by the global curriculum (lines 7-9). If

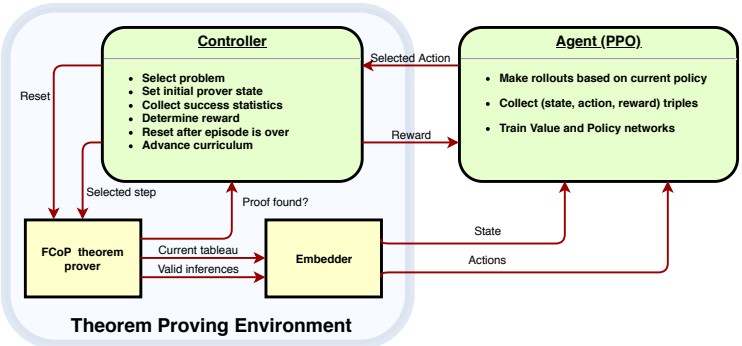

Figure 3: *Theorem proving as a reinforcement learning environment*

we do not have a proof then we start from the beginning. If the policy succeeds in finding a proof of a yet unproven problem then we reset the global curriculum to 1 in line 20. We sample $k$ episodes repeating $k$ times the loop in lines 6-20 and finally decide whether to increase the global curriculum in lines 23-24. We can advance curriculum globally (as in Algorithm 1) or independently for each problem. We found that global advancement makes learning more stable, so that is our default approach. We can start learning with or without training proofs. It does not change the processing of Algorithm 1. In Section 6 we provide experimental evidence with regard to both approaches.

## 5  Datasets

We introduce a suite of problems in the theory of Robinson Arithmetic (Robinson, 1950). This theory is rather limited, however, it seems to be particularly suited for machine learning methods: solutions are long, repetitive and rather predictable

Table 1: *Axioms of Robinson Arithmetic. These are extended automatically with special axioms for handling equality: reflexivity, symmetry, transitivity as well as three congruence axioms (of $s$, $plus$ and $mul$).*

| Name | Axiom |
|------|-------|
| zeroSuccessor | $\forall X : \neg(o = s(X))$ |
| diffSuccessor | $\forall X, Y : (s(X) = s(Y)) \Rightarrow (X = Y)$ |
| addZero | $\forall X : plus(X, o) = X$ |
| addSuccessor | $\forall X, Y : plus(X, s(Y)) = s(plus(X, Y))$ |
| mulZero | $\forall X : mul(X, o) = o$ |
| mulSuccessor | $\forall X, Y : mul(X, s(Y)) = plus(mul(X, Y), X)$ |

for humans. Still, as we shall see in Section 6, state-of-the-art systems struggle to solve some of these problems. Each problem is a separate environment and a guidance system is expected to learn on some of them and then perform well in unseen environments, which is a meta-learning task. The problem sets are intended to be a general purpose testing ground for theorem proving and RL methods, in particular for meta-learning and hierarchical learning algorithms.

Robinson Arithmetic defines basic properties of arithmetic expressions. The signature of the language contains an atom '$o$' (representing 0), functions '$s$', '$plus$' and '$mul$' (representing $+1$, $+$ and $\cdot$,

---

**Algorithm 1** FLoP: Curriculum Learning on Proofs

---

      **Input:** problem set $\mathcal{P}$, policy $\pi$, progress threshold $\in [0..1]$
              train steps $\in \mathbb{N}$, episodes between updates: $k \in \mathbb{N}$
      **Output:** trained policy $\pi$, possible new proofs for problems in $\mathcal{P}$

 1: steps $\leftarrow 0$
 2: *curriculum $\leftarrow 1$*
 3: **while** steps < train steps **do**
 4:     successes $\leftarrow 0$
 5:     **for** j in 1..k **do**
 6:         $p \leftarrow$ random problem from problem set $\mathcal{P}$       ▷ An episode corresponds to a problem
 7:         **if** $p$ has stored proof **then**                 ▷ Determine initial state
 8:             Take proof steps according to stored proof until *curriculum* number of steps remain
 9:             $s_0 \leftarrow$ state of problem $p$ after initial proof steps taken
10:         **else**
11:             $s_0 \leftarrow$ starting state of problem $p$
12:         **while** not episode over **do**
13:             Take action according to policy $\pi(a_i|s_i)$, observe next state $s_{i+1}$ and reward $r_{i+1}$
14:             steps $\leftarrow$ steps + 1
15:         **if** proof is found for $p$ **then**
16:             successes $\leftarrow$ successes + 1
17:             **if** found proof is shorter than previous proof **then**
18:                 store proof as new proof for $p$
19:             **if** no proof of $p$ was known before **then**
20:                 *curriculum $\leftarrow 1$*         ▷ Restart curriculum learning
21:     Update policy $\pi$
22:     success rate $\leftarrow$ successes / k
23:     **if** success rate > progress threshold **then**
24:         *curriculum $\leftarrow$ curriculum + 1*        ▷ Advance curriculum

---

respectively), and the equality predicate '='. For example, formula $3 \cdot 1 + 2 = 4 + 1$ is written as

$$plus(mul(s(s(s(o))), s(o)), s(s(o))) = plus(s(s(s(s(o)))), s(o)).$$

We use the axioms provided in Table 1. The unary representation of numbers (e.g., $s(s(s(o)))$ represents 3) results in large expressions and long proofs as the numbers increase. For example, $((8 + 5) \cdot 8) \cdot 5 = 520$ takes over 16000 steps to prove in fCoP. We show an example of such a proof on the project website. In Table 2 we identify three problem sets of increasing complexity that we use in Section 6 to evaluate FLoP. For Stage 1, a good ordering of the available inference actions is sufficient to find a proof. Stage 2 is harder, as the current goal is also important for selecting an action. For example, the equality reflexivity $A = A$ is usually not needed, except for some cases, due to the unification it triggers. So the system has to learn that this action is useful in particular situations. Stage 3 is much harder, because some of the "rare" actions are tied to global progress in the proof, for example, when we move focus from the left side of the equation to the right side.

## 6 EXPERIMENTS

In this Section we present six experiments. Experiment 1 demonstrates that FLoP can learn when no proof is provided, only the training problems. In Experiment 2 we compare FLoP with other provers and show that it performs very well on the arithmetic datasets. In Experiment 3 we show that FLoP tends to solve problems in fewer steps than rlCoP and also solves problems that require significantly longer proofs. Experiment 4 shows that FLoP can generalize from a single training problem that is provided with a proof. In Experiment 5 we show that proofs of harder problems provide more valuable training signal and we obtain the best generalization when we learn from some longer proofs. Finally Experiment 6 shows that FLoP is more robust than supervised learning on training proofs.

In total we used around 2.5M core-hours on Xeon E5-2680v3 processors, approximately 250-300 core-years. Our hyperparameters were selected using small grid searches. We checked standard

Table 2: *Three challenges defined in the theory of Robinson Arithmetic. The eval set contains all problems for stage 1 and is randomly sampled for stages 2 and 3.*

| Stage | Set | Size | Description |
|---|---|---|---|
| **Stage** 1 | Train | 2 | $1+1=2, 1\cdot 1=1$ |
| | Eval | 1800 | Expressions of the form $N_1 + N_2 = N_3$, $N_1 \cdot N_2 = N_3$, where $0 \leq N_i < 30$. (Examples: $3+4=7$ or $5\cdot 12=60$) |
| **Stage** 2 | Train | 3 | $1+1=2, 1\cdot 1=1, 1\cdot 1\cdot 1=1$ |
| | Eval | 1000 | $T = N$, where $0 \leq N$, and $T$ is a random expression with 3 operators and operands $N_i$ such that $0 \leq N_i < 10$. (E.g.: $((3+4)\cdot 2)+6=20$) |
| **Stage** 3 | Train | 810 | $T_1 = T_2$, where $T_1$ and $T_2$ are random expressions with 3 operators and operands $N_i$ such that $0 \leq N_i < 2$. |
| | Eval | 1000 | $T_1 = T_2$, where $T_1$ and $T_2$ are random expressions with 3 operators and operands $N_i$ such that $2 \leq N_i < 10$. (E.g. $((3+4)\cdot 2)+6=((1+1)\cdot 5)\cdot 2$) |

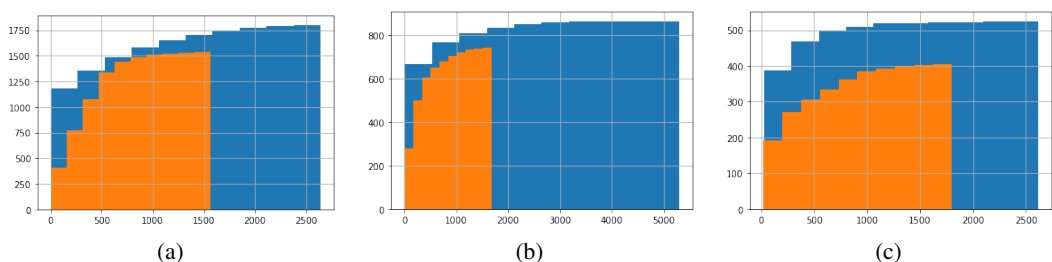

| (a) | (b) | (c) |
|---|---|---|

Figure 4: *Comparing the length of proofs found by FLoP (blue) and rlCoP (orange) on the Robinson Arithmetic dataset. All figures are cumulative histograms, vertical axes show the number of proofs, horizontal axes show the length of proofs. Best models are shown for both FLoP (based on Experiment 3) and rlCoP. Figures (a), (b), (c) correspond to Stage 1, 2, 3 respectively. FLoP found more proofs in all stages.*

RL parameters (e.g., the discount factor) parameters related to curriculum scheduling (e.g., local vs. global), neural network architectures (1–5 layers with 128–1024 neurons), feature sizes (64–1024) and training steps ($10^5 - 10^8$). Parameters used in the experiments are described in configuration files which are accessible along with the shared codebase.

Each model was trained to achieve 100% accuracy on the training set. During evaluation, the system was allowed to make 100 attempts per problem, each with 60 sec. time limit, without backtracking. We report two evaluation metrics: 1) *Succ.*: percentage of proofs found and 2) *Len.*: average length of proofs found. We have trained 5 models per a set of hyperparameters (unless otherwise noted). Reported numbers are means, with standard deviations in parenthesis. We discuss some failure modes in Appendix A.

**Experiment 1: Learning without proofs.** We train FLoP with the training problems defined in Section 5, without proofs. The system can find training proofs through exploration and learn from them as we show in Table 3. Curriculum learning performs well for Stage 1 and 2, however in Stage 3 it only solves 3%. This massive overfitting, addressed in Experi-

Table 3: *Curriculum learning compared with only exploration based learning.*

| | No curriculum | | Curriculum | |
|---|---|---|---|---|
| Stage | Succ. | Len. | Succ. | Len. |
| 1 | **1.0** (0.0) | 361(2) | **1.0** (0.0) | 359(3) |
| 2 | 0.46(0.09) | 88(36) | **0.75** (0.08) | 255(93) |
| 3 | **0.47** (0.05) | 273(27) | 0.03(0.03) | 289(183) |

ments 4 and 5, happens because the system tends to overuse equality congruence axioms, i.e. when trying to prove $a + b = c + d$ or $a \cdot b = c \cdot d$, it often proceeds by reducing the problem to proving $a = c$ and $b = d$, which does not work in general. In the training set, all numbers are 0 and 1 and this approach works more often. We also compare curriculum learning with learning based on exploration only. As Figure 5 in the Appendix shows, curriculum learning yields more rewards. This makes no difference in the setting of Stage 1, helps greatly in Stage 2 and results in overfitting in Stage 3.

**Experiment 2: Comparison with other Provers.** We compare FLoP with two state-of-the-art saturation-style theorem provers (E, Vampire), a strong connection tableau prover (leanCoP (Otten &

Bibel, 2003)) and one connection tableau prover using learning-based guidance (rlCoP (Kaliszyk et al., 2018)).

Vampire, E and leanCoP use human-designed strategies instead of learning. In our tests we use the *casc* mode for Vampire, the *auto* and *auto-schedule* modes for E and the default collection of 40 strategies for leanCoP (the standard casc setup), each with a timeout of 60 sec. per problem. For rlCoP we used the same hyperpa-

Table 4: *Comparing Vampire, E, leanCoP, rlCoP and FLoP, with respect to success ratio for Stage 1, 2 and 3 problems. Our method (FLoP) is marked in grey. $E_1$ – auto mode, $E_2$ – auto-schedule mode.*

| Stage | Vampire | $E_1$ | $E_2$ | leanCoP | rlCoP | FLoP |
|-------|---------|-------|-------|---------|-------|------|
| 1 | 0.60 | 0.60 | **1.0** | 0.22 | 0.86 | **1.0** |
| 2 | 0.40 | 0.39 | **1.0** | 0.14 | 0.74 | 0.84 |
| 3 | 0.34 | 0.28 | **1.0** | 0.01 | 0.41 | 0.51 |

rameters as those described in Kaliszyk et al. (2018), only modifying the policy temperature from 2.5 to 1.5. The number of inferences in the MCTS was limited to 200000. For Stage 1 and 2 rlCoP was run directly on the evaluation set. For Stage 3 all problems were too hard to solve without guidance within the inference limit, so we started with the version trained on the solutions of Stage 2. For FLoP we report the best models trained without proofs.

The success ratios are given in Table 4. E's *auto-schedule* tries multiple strategies and finds one with the left-to-right ordering of all the addition and multiplication axioms. This solves all of our problems immediately without any proof search by only using rewriting to a normal form (Baader & Nipkow, 1998). This demonstrates the power of equational theorem proving when a suitable term ordering exists and can be found by human-designed heuristics. This is however far from guaranteed in general and nontrivial even in such simple domains, as witnessed by Vampire's failure to find this ordering. To evaluate E without access to its built-in rewriting capability, we have renamed the equality to a new predicate 'eq' axiomatized exactly in the same way as in fCoP. The auto-schedule mode then solves $54\%$ problems in Stage 1, comparable to the auto mode. Overall, FLoP solves the most problems in all stages if we count systems that rely on search space exploration.

**Experiment 3: FLoP vs. rlCoP with Respect to Proof Lengths.** Due to the same underlying calculus, the proofs found by rlCoP and FLoP are directly comparable and it is insightful to compare them with respect to the length

Table 5: *Comparing rlCoP and FLoP with respect to proofs found and proof lengths on Stage 3.*

|  | found | found only | found by both | shorter proof |
|------|-------|-----------|---------------|---------------|
| rlCoP | 406 | 55 | 351 | 53 |
| FLoP | **517** | **166** | 351 | **298** |

of proofs. Figure 4 shows that FLoP manages to solve more problems, and even finds some very long proofs. This is, however, not because FLoP's proofs are unnecessarily long: we demonstrate in Table 5 that FLoP tends to find shorter proofs for problems solved by both systems. It is interesting to note that out of the 351 problems solved by both, none had the same length, which suggests that the provers acquired different strategies.

**Experiment 4: Learning from a Single Problem with Proof Provided.** When the proof of training problems is available, FLoP can use it to make learning more efficient. In this experiment, FLoP is restricted to learn from a single, rather simple training problem, but we also provide its

Table 6: *Curriculum Learning on a single training proof. Numbers with # and ⋆ are based on 1, 2 runs, respectively.*

| Stage | Training problem | Succ. | Len. |
|-------|-----------------|-------|------|
| 1 | $1 \cdot 1 = 1$ | 1.0(0) | 368(5) |
| 2 | $1 \cdot 1 \cdot 1 = 1$ | 0.71(0.01)⋆ | 311(61)⋆ |
| 3 | $1 \cdot 1 \cdot 1 + 1 = 1 \cdot 1 + 0 + 1$ | 0.37# | 336# |

proof. Table 6 shows that FLoP generalizes very well in this setup, expecially in Stage 1 and 2.

Table 7: *Curriculum learning for Stage 3 on two harder problems with proofs of 113 and 108 steps. Results are based on 3 runs.*

| Training problem | Succ. | Len. |
|-----------------|-------|------|
| $1 \cdot 2 + 1 + 1 = (1+1) \cdot 1 \cdot 2$ | 0.32(0.05) | 566(14) |
| $1 \cdot 2 + 1 + 1 = (1+1) \cdot 1 \cdot 2$ $(1+1+1) \cdot 2 = 2 \cdot 1 + 2 + 2$ | **0.51** (0.03) | 590(54) |

**Experiment 5: Learning from long proofs.** The massive overfitting in Stage 3 shown in Figure 5 is endemic to modern RL methods (Zhang et al., 2018). Here, in particular, the system tends to overuse equality congruence axioms, i.e. when trying to prove $a + b = c + d$ or $a \cdot b = c \cdot d$, it often proceeds by reducing the problem to proving $a = c$ and $b = d$, which does not work in general. In the training set, all numbers are 0 and 1 and this approach works more often. The harder the problems, the less likely they can be solved with such heuristic approaches, hence harder training problems promise more valuable training signal. We demonstrate this by training FLoP on a few selected harder problems with proofs provided. A single longer training proof (113 steps) is sufficient to discourage the overuse of equality axioms. Adding one more training problem (108 steps) helps even more and

we obtain the best model for Stage 3, see Table 7. Also note the huge increase in the length of proofs: this is partly because we solve new hard problems and partly because the system resorts to longer but safer proof strategies.

Table 8: *Curriculum Learning vs Supervised Learning on Stage* 1 *and* 2*, using training proofs with some* 1 *-* 3 *steps added for distraction.* FLoP *is barely affected, while supervised learning's performance degrades. Numbers with* $\star$ *are averaged from 2 runs.*

| Stage | Proof Lengths | Supervised | | Curriculum | |
|---|---|---|---|---|---|
| | | Succ. | Len. | Succ. | Len. |
| 1 | 5, 9 | 0.98(0.04) | 327(58) | **1 (0.01)** | 363(5) |
| 1 | 7, 10 | **1 (0)** | 359 (0) | 0.98(0.01) | 327(18) |
| 1 | 9, 11 | 0.52(0.08) | 54(11) | **0.98 (0.01)** | 340(18) |
| 2 | 5, 9, 23 | **0.85 (0.04)** | 377(47) | 0.76(0.02)$^\star$ | 291(16)$^\star$ |
| 2 | 7, 10, 24 | **0.74 (0.04)** | 433(110) | 0.71(0.01)$^\star$ | 311(61)$^\star$ |
| 2 | 9, 11, 25 | 0.59(0.08) | 193(49) | **0.76 (0.01)**$^\star$ | 267(109)$^\star$ |

**Experiment 6: Curriculum Learning vs Supervised Learning.** When training proofs are available, a natural baseline of curriculum learning is supervised learning on the proof steps. While such behavioral cloning sometimes leads to great performance, we show in Table 8 that it greatly depends on the quality of the given proof. For the three problems in the training set of Stage 1 and 2, we take the shortest proofs (5, 9 and 23 steps) and construct variants with 1-3 extra steps added. We observe that supervised learning degrades as superfluous steps are introduced, while FLoP's exploration allows the system to recover and find the original proofs.

## 7 CONCLUSION AND FUTURE WORK

We have built FLoP, a proof guidance system based on reinforcement learning addressing the problem of finding long proofs in an exponential search space. Previous work (Kinyon et al., 2013; Veroff, 1996) focused on finding long proofs with the help of human-designed heuristics. We find that curriculum learning is a suitable approach as it strikes a good balance between exploration and memorization on a suite of arithmetic RL problems introduced in Section 5, allowing our system to generalize from small training problems to larger ones with similar structure. We have created a suite of RL environments based on Robinson arithmetic that contains problems of highly related structure.

Overfitting in Reinforcement Learning is a a well known and largely unsolved problem and we believe that our work offers an interesting new angle on the problem. We show that the greater reward efficiency provided by curriculum learning can result in catastrophic overfitting. Figure 5 and Table 3 show that allowing or disallowing the curriculum can be considered as a trade-off between more efficient training and a higher risk of overfitting. Furthermore, we also show in Table 7 that when we have some training proofs of harder problems, we can greatly reduce overfitting.

This work uses a human-designed representation of formulas similar to one used earlier in Kaliszyk & Urban (2015b); Kaliszyk et al. (2018); Urban et al. (2011) and a straightforward encoding of actions. We believe that learned embeddings as well as more sophisticated RL ideas employed before in the context of text games (Côté et al., 2018; Fulda et al., 2017; Haroush et al., 2018; He et al., 2015; Kostka et al., 2017; Narasimhan et al., 2015) will positively impact the performance of FLoP. We also see potential in exploring other ways of curriculum learning: while in this paper curriculum is on the number of steps to the end of the proof, one could order training problems from easier to more complex ones.

We believe that the learning loop implemented in Algorithm 1 can benefit from integration with memory-based meta-learning methods (Ortega et al., 2019; Santoro et al., 2016). One can also look for inspiration from robotics (Florensa et al., 2017) with regard to automatic discovery of curricula of easier problems. In mathematical practice it is quite often the case that instead of proving a conjecture, a similar, but simpler problem is solved, which eventually – maybe over a longer span of time – contributes to the solution of the main conjecture. This encourages us to combine FLoP with Hindsight Experience Replay (Andrychowicz et al., 2017), which utilizes a similar strategy of allowing easier goals in order to ultimately solve a harder goal in the domain of robotics.

Finally we find it interesting to instrument the Bolzano-Weierstrass theorem and its 252 auxiliary lemmas as an RL challenge where the system would be supposed to derive the theorem and all lemmas from scratch using in each derivation only basic axioms, hence forcing long proofs. This can be considered as an RL follow-up to the earlier MPTP Challenge (Urban, 2006b).

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

## APPENDIX A    FAILURE MODES

Despite the apparent simplicity of our arithmetic learning environments, a learning system aiming to solve them has to overcome some hard challenges. We have decided to describe these challenges in detail as they are present in other domains as well, even if it may be harder to detect.

**Failure type 1.** The reward mechanism of our RL system is biased towards shorter proofs. However, many problems have "shortcuts" that allow for shorter proofs, but that do not generalize well. Consider formula $(1 + 1) + (2 \cdot 2) = (0 + 2) + 4$. There are two ways to prove this equality: 1) compute the values of the expressions on both sides of the equation and notice that they are the same or 2) show that $1 + 1 = 0 + 2$ and $2 \cdot 2 = 4$. The former generalizes better, but the latter results in a shorter proof. Hence, training on this problem might negatively affect the performance of the prover. This is what causes the failure in Experiment 3: through manual inspections of discovered proofs we have concluded that curriculum learning is more efficient at finding and learning shorter proofs of the training problems and it overfits to them.

**Failure mode 2.** fCoP features do not take into account the order of the arguments of a function, hence $f(a, b)$ and $f(b, a)$ have the same features. This is particularly problematic for Stage 3, since $A = B$ and $B = A$ require different inferences. We addressed this problem by 1) extending state features with those of the preceding action as a substitute of a memory, 2) modified the features to include argument order.

**Failure mode 3.** Some "rare" events are hard to generalize, because the system sees very few relevant samples during training. This is the case with applying commutativity of equality (replacing $A = B$ with $B = A$), which is only required in Stage 3 and ideally only once per proof, when we move focus from one side of the equation to the other. In Experiment 4, when we trained on a single longer proof, we have noticed that the system was very unsure about this action which resulted in many failed proof attempts. Adding another training proof as enough to overcome this and success score increased from 32% to 51%.

## APPENDIX B    THE EFFECT OF CURRICULUM LEARNING

Figure 5 shows that curriculum learning yields more rewards. This makes no difference in the simple setting of Stage 1, helps greatly in Stage 2 and results in fatal overfitting in Stage 3.

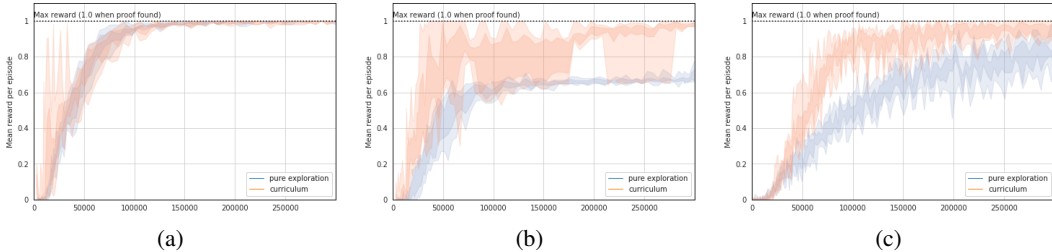

|     (a)     |     (b)     |     (c)     |

Figure 5: *(a)-(c) – Stages 1-3, training graphs centered at the mean reward, darker bars are delimited by quantiles at 0.25 and 0.75, lighter bars extending from min to max; in total 36 models, 6 models per graph, 20M samples per experiment. Curriculum helps in Stages 2 and 3.*

