# OpenReview forum: "Towards Finding Longer Proofs"
_ICLR.cc/2020/Conference — Reject_

### Official Review · AnonReviewer1 · 2019-10-08
**Official Blind Review #1**

**Rating:** 6

**Review:**

OVERALL:

I don't work on ATP and am not particularly well suited to review this paper, but I am
slightly inclined to accept for the following reasons.

1. It's an important problem and not much work is done on it.
2. Creating new environments is hard, valuable work that is (IMO) insufficiently incentivized in our community,
and I think accepting papers that do this work is a good policy.
3. The experiments, to my super-inexperienced eye, seem well designed and like they address
obvious questions readers would have.

However, I have no idea if e.g. the baselines used in this paper are reasonable, so
I would appreciate someone with more experience on that topic weighing in.


Some limitations of the paper:

1. I'm not actually convinced there's much that's methodologically new here.
It seems like mostly an application of existing RL techniques to an ATP environment.

2. The writing is not particularly clear, and could use substantial editing (but this
is something that could be fixed during the discussion period).

3. The focus on Robinson arithmetic seems kind of limiting.
Though I am sympathetic to the reasoning given in the paper, it's unclear to me (again, as a non-expert),
that the techniques that work in this context will actually work in the context considered by e.g. [1]

DETAILED COMMENTS:
> In the training set, all numbers are 0 and 1 and this approach works more often.
You have this sentence twice in different places.

> it is insightful to compare...
Not a very idiomatic use of insightful?


[1] Deep network guided proof search.


**Experience Assessment:**

I do not know much about this area.

**Review Assessment: Checking Correctness Of Derivations And Theory:**

N/A

**Review Assessment: Checking Correctness Of Experiments:**

I assessed the sensibility of the experiments.

**Review Assessment: Thoroughness In Paper Reading:**

I read the paper at least twice and used my best judgement in assessing the paper.

---

> ### Author Response · Authors · 2019-11-13
> **Answer to Reviewer #1**
>
> Dear Reviewer,
> Thank you for your comments.
>
> You are right that from a strictly RL perspective, our system brings no methodological novelty. However, this is a new method in automatic theorem proving and we argue that it is a good approach to address the sparse reward problem of theorem proving, a problem that hasn't been tackled before.
>
> With respect to your concern about the simplicity of the Robinson Arithmetic dataset, please note that the dataset is already hard for state of the art theorem provers, as shown in Figure 4. In order to solve these simple problems, you have to manually calibrate a human heuristic, otherwise known proof methods fail as the proofs to be found get longer. Furthermore, this dataset allowed us to identify some important failure modes that are not particular to Robinson Arithmetic or to FLoP. We claim that our datasets are of the very few that not only challenge theorem provers, but also allow for understanding their current limits. Please see more about the dataset in the answer given to Reviewer #3.

---

### Official Review · AnonReviewer2 · 2019-10-21
**Official Blind Review #2**

**Rating:** 3

**Review:**

This paper uses reinforcement learning for automated theorem proving. The proposed method aims to generalize the short proofs to longer proofs with similar structure. Experiments were run to compare the performance of curriculum learning with the ones without curriculum.

Overall the paper attempts to explain clearly the original contribution of the proposed approach, which is using curriculum learning in RL based proof guidance. However,  I am not convinced about the how compelling the results are in support of the claim. The main arguments to bolster my decision are as follows.

I am familiar with RL and curriculum learning but not so much with connection tableau calculus. The description given in the paper seems to be insufficient and confusing for readers with limited knowledge in this area. A step-by-step explanation with a toy example might have done the job nicely. Without such a clear understanding of the calculus, it gets hard to appreciate the merits of the results.

Some claims of the paper are not clearly validated by the reported experimental results. For example:
- in Table 8, curriculum learning is worse in some cases and better in others. What to conclude from such a report?
- in experiment 3, curriculum learning tends to find shorter proofs. Isn't that contrary to the focus of the paper?
- in Table 3, curriculum learning performs lot worse than the other method. What is to be inferred from such a report?

It would be nice if there was a clear explanation of the role of curriculum in the learning algorithm. For example, in Algorithm 1, how is Line 8 helping in overall objective of learning longer proofs? If one advances curriculum, one takes lesser number of proof steps according to stored proofs. How does that help in the learning? Does it imply 'less memorizing' with advancement of curriculum?

**Experience Assessment:**

I have read many papers in this area.

**Review Assessment: Checking Correctness Of Derivations And Theory:**

I carefully checked the derivations and theory.

**Review Assessment: Checking Correctness Of Experiments:**

I assessed the sensibility of the experiments.

**Review Assessment: Thoroughness In Paper Reading:**

I read the paper at least twice and used my best judgement in assessing the paper.

---

> ### Author Response · Authors · 2019-11-13
> **Answer to Reviewer #2**
>
> Dear Reviewer,
> Thank you for your comments.
>
> It is a challenging task to provide a proper introduction to the connection calculus in such a short paper. However, we did our best to provide illustration on the project webpage http://bit.ly/site_atpcurr . Here you can find some screencasts and logs that cover few selected problems and include all details of the reasoning.
>
> The primary role of our curriculum is to provide more training signal to the learner, as well as to allow a priori knowledge to the system through proofs (please see the answer given to Rewiever #3). Figure 5 in Appendix B illustrates that curriculum learning indeed yields more reward during training. The upside is that this makes training faster and more stable. However, the downside of better is training is the risk of overfitting. We can see our curriculum as a tradeoff between faster training and higher chance of overfitting. As described in Appendix A, Failure Modes, Stage 3 is very sensitive to overfitting as different proofs of the same problem can yield very different generalization, this is why curriculum can be detrimental.
>
> RL methods are oftenly biased towards shorter solutions: even if no reward discounting is applied, exploration is more likely to find shorter solutions than longer ones. This is true with and without curriculum. What curriculum learning does is to speed up learning once a proof was found. So it is beneficial if used on some "good" proofs and detrimental if used on "bad" proofs. Table 3 shows an example of each scenario (Stage 2 and 3). Assessing the quality of proofs (with respect to generalization) is, we believe, an important research question.
>
> While our project targets to learn to solve hard problems that require long proofs, given a particular problem, we have no incentive to prefer longer proofs of that problem. Our only concern during training is to learn how to generalize to other problems.
>
> The advancement of curriculum is meant to ensure that the system is continuously faced with a "reasonably" hard problem. Once it becomes easy to finish the proof from a particular state, we make things harder by moving the starting state backwards. It is only after we have gone through the full curriculum that the system has been exposed to all the states of the proof.
>
> The reason why we believe our setting is useful for learning long proofs is that we are capable of performing very long rollouts during training time, without facing an exponentially diminishing chance of getting some reward. The system thus trains on long sequences of proof steps.

---

### Official Review · AnonReviewer3 · 2019-10-23
**Official Blind Review #3**

**Rating:** 3

**Review:**

In the paper, the authors present a new algorithm for training neural networks used in an automated theorem prover using theorems with or without proofs as training data. The algorithm casts this training task as a reinforcement learning problem, and employs curriculum learning and the Proximal Policy Optimization algorithm to find appropriate neural network parameters, in particular, those that make the prover good at finding long proofs. The authors also propose a new dataset for theorems and proofs for a simple equational theory of arithmetic, which is again suitable for improving (via learning) and testing the ability of the prover for finding long proofs. The proposed prover is tested against existing theorem provers, and for the authors' dataset, it outperforms those provers.

I found it difficult to make up my mind on this paper. On the one hand, the paper tackles an interesting problem of improving an automated theorem prover via learning, in particular, its ability for finding long nontrivial proofs. Also, I liked a qualitative analysis of the failure of the curriculum learning for tackling hard tasks in the paper. On the other hand, I couldn't quite make me excited with the dataset used to test the prover in the paper. The dataset seems to consist of easy variable-free equational formulas about arithmetic that can be proved by evaluation. Of course, I may be completely wrong about the value of the dataset. Also, if the dataset includes variables and other propositional logic formulas, such as disjunction, negation and conjunction, so that the prover can be applied to any formulas from Peano arithmetic via Skolemization, I would be much more supportive for the paper. Another thing that demotivated me is that I couldn't find the discussion about the subtleties in using curriculum learning and PPO for the theorem-proving task in the paper. What are the possible design choices? Why does the authors' choice work better than others?

I added a few minor comments below.

* abstract, p1: "significantly outperforms previous learning-based". When I read the experimental result section, I couldn't quite get this sense of huge improvement of the proposed approach over the existing provers. Specifically, from Table 4, I can see FLoP performs better than rlCoP, but I wasn't sure that the improvement was that significant (especially because rlCoP might not have given a chance to be tuned to the type of questions used to train FLoP -- I may be wrong here). I suggest you to add some further explanation so that a reader can share your sentiment and excitement on the improvement brought by your technique.

* p2: The related work section is great. I learned a lot by reading it. Thanks.

* p4: I think that you used the latex citation command incorrectly in "learning Resnick ... Chen (2018)"
and "features Kaliszyk ... Kaliszyk et al. (2015a; 2018)".

* p6: discount factor) parameters related ===> discount factor), parameters related

* p8: a a well ==> a well



**Experience Assessment:**

I have published one or two papers in this area.

**Review Assessment: Checking Correctness Of Derivations And Theory:**

N/A

**Review Assessment: Checking Correctness Of Experiments:**

I assessed the sensibility of the experiments.

**Review Assessment: Thoroughness In Paper Reading:**

I read the paper thoroughly.

---

> ### Author Response · Authors · 2019-11-13
> **Answer to Reviewer #3**
>
> Dear reviewer,
> Thank you for your comments.
>
> You raised concerns about the usefulness of our arithmetic datasets. Please keep in mind, that despite the simplicity, these problems are already challenging for state of the art automated theorem provers, as shown in Table 4. Something that works here might not necessarily work directly on more complex datasets, but we argue that the problems that we observe when trying to solve these simple arithmetic equations are equally present elsewere, even if not so easily observable. Nevertheless, we agree that the way forward is to
> target more and more complex datasets.
>
> We were able to identify the dominant failure modes, that are mostly related to overfitting to the training problems and more specifically to some particular proofs of the training problems. We believe that we are the first to raise the issue that valid proofs of the same problem might differ greatly in terms of generalizability. We haven't yet found a satisfactory solution to this problem, but we still think this is an important message to the community. These failure modes are not particular to Robinson Arithmetic or FLoP.
>
> In order to appreciate the experimental results, it is useful to consider the lengths of proofs found by the various provers. Table 5 and Figure 4 together show that FLoP tends to find shorter proofs of the same problem than rlCoP,
> still the former finds much more longer proofs, suggesting that it solves more of the harder problems. We can also look at lengths of proofs found by the other (unguided) provers. Even if proofs of the same problem cannot be
> compared directly, due to the different underlying calculus, it is still an important question how well one prover can solve problems that require long inference chains. If we omit proofs in Stage 3 that are less than 100 steps long, then the number of proofs found by the provers is: Vampire (107), E (0 - including variants with auto and autoschedule mode, as well as the one where equation is replaced with the eq symbol), rlCoP (340), FLoP (481). This
> suggests that the advantage that FLoP obtained over its peers is in the range of harder problems, requiring several hundred proof steps. Also note that leanCoP is a rather simple and weak prover (as shown in Table 4), still guidance has allowed it to obtain competitive results.
>
> Curriculum learning has not yet previously been applied to theorem proving and we don't claim that our work is the only/best way to do it. We argue, however, that this will be an important component of provers that approach human level competence. Our main design choice here was to do curriculum on the length of a proof (whether given from outside or found by the prover). Another natural option for curriculum is to order problems according to their difficulty and first learn from the easier ones. While this is easy to implement in simple theories like Robinson Arithmetic, assessing the difficulty of a problem is a hard task in general.
>
>
> Our approach to curriculum has two important benefits:
> - We alleviate the sparse reward problem of theorem proving. More reward signal results in faster and more stable training.
> - We provide a comfortable mechanism to incorporate prior knowledge into the system in the form of proofs. As Table 7 shows, our best results were obtained when the system was given a couple longer proofs, so it didn't have to bootstrap from scratch. There can be theories where this is the only option to start the system because even the easiest problems are too hard to find through pure exploration. In FLoP, it is now a design choice of the user what kind of proofs to start with.
>
> This curriculum is independent from PPO, we could equally use any other popular RL method. We just make it easier for an episode to terminate in a reward yielding state.

---

> > ### Comment · AnonReviewer3 · 2019-11-14
> > **Thanks!**
> >
> > Your response was very helpful. Thanks!

---

### Decision · Program_Chairs · 2019-12-19

**Decision:**

Reject

**Comment:**

This paper proposes a curriculum-based reinforcement learning approach to improve theorem proving towards longer proofs. While the authors are tackling an important problem, and their method appears to work on the environment it was tested in, the reviewers found the experimental section too narrow and not convincing enough. In particular, the authors are encouraged to apply their methods to more complex domains beyond Robinson arithmetic. It would also be helpful to get a more in depth analysis of the role of the curriculum. The discussion period did not lead to improvements in the reviewers’ scores, hence I recommend that this paper is rejected at this time.